# The Role of Adrenoceptors in the Retina

**DOI:** 10.3390/cells9122594

**Published:** 2020-12-03

**Authors:** Yue Ruan, Tobias Böhmer, Subao Jiang, Adrian Gericke

**Affiliations:** 1Department of Ophthalmology, University Medical Center, Johannes Gutenberg University Mainz, Langenbeckstr. 1, 55131 Mainz, Germany; tboehmer@joho-rheingau.de (T.B.); sjiang@uni-mainz.de (S.J.); 2St. Josef’s Hospital Rheingau, Eibinger Str. 9, 65385 Rüdesheim am Rhein, Germany

**Keywords:** α_1_-AR, α_2_-AR, β-AR, retina, distribution, function

## Abstract

The retina is a part of the central nervous system, a thin multilayer with neuronal lamination, responsible for detecting, preprocessing, and sending visual information to the brain. Many retinal diseases are characterized by hemodynamic perturbations and neurodegeneration leading to vision loss and reduced quality of life. Since catecholamines and respective bindings sites have been characterized in the retina, we systematically reviewed the literature with regard to retinal expression, distribution and function of alpha_1_ (α_1_)-, alpha_2_ (α_2_)-, and beta (β)-adrenoceptors (ARs). Moreover, we discuss the role of the individual adrenoceptors as targets for the treatment of retinal diseases.

## 1. Introduction

Many ocular diseases, such as age-related macular degeneration (AMD), diabetic retinopathy, retinal vascular occlusion and glaucoma frequently cause severe visual impairment substantially diminishing the quality of life [1,2,3,4,5]. Permanent ischemia or ischemia-reperfusion injury of the retina and/or optic nerve have been implicated in the pathophysiology of these diseases [6,7,8,9]. Retinal ischemic conditions may lead to a direct loss of cellular function. However, they can also induce various secondary sequelae, such as retinal edema, hemorrhage, neovascularization, detachment and secondary glaucoma [10].

Since the retina is the tissue with the highest metabolic demand in the body, it is not surprising that retinal hemodynamic perturbations play a critical role in the pathogenesis of numerous ocular diseases [11,12,13,14,15,16,17,18]. The retinal vascular structure provides metabolic support for neural and glial cells while minimizing interference with light-sensing [19]. In this context, modulation of vascular diameter by local mechanical and chemical stimuli can be considered the main basic regulatory mechanism in the retinal circulation [20,21].

Receptor families comprising multiple subtypes with organ-specific patterns of neurovascular expression and function appear to be attractive candidates for the development of targeted therapies against diseases associated with abnormal tissue perfusion. One such intriguing receptor family is the adrenoceptor (AR) family.

The discovery of ARs by Ahlquist more than six decades ago has uncovered the adrenergic signaling pathways as pivotal regulators of systemic arterial blood pressure and other metabolic and central nervous system functions [22]. ARs belong to the superfamily of guanosine triphosphate-binding protein (G protein)-coupled receptors (GPCRs) and are targets of catecholamines, especially epinephrine and noradrenaline [22,23,24,25]. According to its pharmacological properties, amino acid sequence, and signaling mechanisms, the AR family is subdivided into three subfamilies, the alpha_1_ (α_1_)-, alpha_2_ (α_2_)-, and the beta (β)-AR subfamily [26]. Each class is composed of three subtypes (α_1A_, α_1B_, α_1D_, α_2A_, α_2B_, α_2C_, β_1_, β_2_, and β_3_) (Figure 1) [27]. Members of the α_1_-AR subfamily are widely expressed throughout the cardiovascular system [28]. They critically participate in the regulation of vascular tone and blood flow primarily by mediating the vasoconstrictive effects of catecholamines [28,29,30,31,32,33,34,35]. Remarkably, the expression pattern of individual α_1_-AR subtypes and their involvement in mediating vascular responses to catecholamines differ considerably between individual vascular beds [34,36,37,38,39]. Members of the α_2_-AR subfamily are expressed in both the central and the peripheral nervous system throughout the body. They are localized either pre- or postsynaptically and mediate inhibition of neurotransmitter release [40,41]. β-ARs are widely distributed at both the central and peripheral nervous system and are involved in important functions activated by circulating catecholamines, such as heart rate regulation, vasorelaxation, bronchodilation, and neurotransmitter release [42].

The purpose of the present review is to summarize the current state of research on the role of α_1_-, α_2_- and β-ARs in the retina. We provide an overview of the expression, structural distribution, and regulation of the individual AR subfamilies and their subtypes in the retina and discuss their therapeutic implications.

## 2. The Retina

The retina is a thin multilayer with neuronal lamination responsible for detecting, preprocessing, and sending visual information to the brain [45]. The neuronal lamination in the retina includes neural circuits containing six major types of neuronal cells: retinal ganglion cells (RGCs), amacrine cells, bipolar cells, horizontal cells, and the cone and rod photoreceptors [46]. Despite the retina’s peripheral location, retinal neurons utilize the same types of neurotransmitters (noradrenaline, dopamine, and acetylcholine) as those of the central nervous system [47]. Since visual formation highly depends on the complex neuronal structure of the retina, a variety of detrimental factors, such as ischemia and oxidative stress, may result in the deterioration of retinal cell function and consequently lead to retinal diseases [48].

Two different circulatory systems are originating from the ophthalmic artery, the retinal circulation and the choroidal circulation, which both supply oxygen and nutrients to the retina [19]. While choroidal blood vessels are innervated and modulated by the autonomic nervous system, no sympathetic nerve fiber terminals have been found in or on the wall of human retinal blood vessels, suggesting that the retinal circulation lacks autonomic innervation [49]. Notably, retinal arterioles were shown to respond to local chemical factors, such as oxygen (O_2_), carbon dioxide (CO_2_), nitric oxide (NO), and hydrogen sulfide (H_2_S) [50,51,52]. Although no evidence for sympathetic nerve fibers has been provided in the retina, catecholamines, including noradrenaline, adrenaline and dopamine, have been detected [53]. One possible source of noradrenaline in the mammalian retina are sympathetic nerve terminals located in the choroid [54]. In support of this concept, a decrease in noradrenaline concentration has been observed in the retina after removal of the superior cervical ganglion, which provides sympathetic input to the choroid [53]. This finding indicates that noradrenaline may originate from sympathetic nerve terminals in the choroidal circulation and reach the ARs in the retina by paracrine diffusion [55].

Dopaminergic amacrine (DA) cells, a class of retinal neurons, synthetize and release dopamine, which is the prevailing catecholamine in the retina [56,57]. In the bovine retina, noradrenaline was detected in the inner nuclear and plexiform layers, and Osborne suggested that retinal tissue can metabolize dopamine to form noradrenaline [58]. Also, amacrine cells have been detected in the retina, which differs morphologically from DA cells and can be detected only after a preload with exogenous noradrenaline [54,59]. It has been proposed that these cells contain only small amounts of endogenous catecholamines, but are equipped with high-affinity uptake properties for exogenous catecholamines [54]. There is also compelling evidence indicating that in addition to noradrenaline, dopamine can also activate α_1_-, α_2_-, and β-ARs, but may be less potent [60]. Importantly, members of all three AR subfamilies, α_1_-, α_2_-, and β-ARs, have been detected in retinal tissue, including blood vessels [61].

## 3. Expression and Function of ARs in the Retina

### 3.1. α_1_-ARs

#### 3.1.1. Expression of α_1_-ARs in the Retina

In contrast to choroidal blood vessels, the retinal vasculature seems to lack autonomic, i.e., adrenergic, cholinergic or peptidergic innervation [49,50,62,63,64]. However, α_1_-ARs have been found in retinal tissue of various mammalian species [65,66]. Particularly with regard to the intra-retinal vasculature, Forster et al. have demonstrated the presence of α_1_-adrenergic binding sites in homogenates of isolated bovine retinal arteries and veins, which were few in number, but of high agonist affinity [61]. In homogenates of isolated murine retinal arterioles, our laboratory has found mRNA for all three α_1_-AR subtypes, which were expressed at similar levels [67].

Several studies provided evidence that the cellular components of the three main layers of a vascular wall, i.e., endothelial cells of the intima, smooth muscle cells of the media, and fibroblasts of the adventitia, have the potential to express α_1_-ARs [29,30,31,32,33]. Furthermore, evidence has been provided that individual α_1_-ARs may exert distinct functions within one vessel [32,68,69]. For example, it has been suggested that in coronary arteries, endothelial α_1B_-ARs mediate vasodilation by stimulating endothelial nitric oxide synthase, while α_1D_-ARs localized on smooth muscle cells mediate vasoconstriction [68].

The cellular location of α_1_-ARs within the architecture of retinal blood vessels is still elusive but may have relevant pharmacological and pathophysiological implications. For example, with regard to α_1_-ARs located on retinal vascular smooth muscle cells, one would expect that the intact blood-retinal barrier prevents circulating hydrophilic molecules, such as catecholamines, from reaching these binding sites [70]. Conversely, they might become well-accessible in pathological states associated with an impaired endothelial barrier function [61,71]. Since a series of enzymes involved in catecholamine synthesis has been localized in retinal cells of different species, retinal vascular α_1_-ARs may also be a target of agonists released from surrounding retinal cells [72,73,74,75]. From a therapeutic point of view, the blood-retinal barrier may be circumvented by administering α_1_-adrenergic ligands into the vitreous body [76].

In addition to retinal vessels, Zarbin et al. localized α_1_-ARs in the outer plexiform layer of the rat retina in vitro by semiquantitative autoradiography using [^3^H]prazosin. The authors reported that α_1_-adrenergic binding sites were only enriched in the outer plexiform layer [77]. Other research groups demonstrated that α_1_-ARs are expressed on retinal pigment epithelium (RPE) of the rabbit and bovine retina, where they modulate K^+^ and Cl^−^ transport and electrical currents [78,79]. Unfortunately, at present, antibody-based methods to localize α_1_-AR subtypes within the tissue structure appear to lack validity, since a growing body of evidence indicates that many commercially available antibodies directed against individual α_1_-AR subtypes do not specifically detect their alleged target antigen [80,81,82,83]. A combined approach of ligand-receptor binding techniques and immunostainings and/or functional studies in tissues from knockout (KO) animal models lacking the respective receptor subtype may represent a more suitable methodological alternative to determine the location and function of α_1_-ARs within the retina, which remains a subject of further research [28].

#### 3.1.2. α_1_-ARs in Retinal Vascular Reactivity

In vivo studies performed in experimental animals and healthy humans have investigated the impact of systemically administered α_1_-adrenergic agonists on retinal vessel reactivity. The results, however, are partly contradictory, which makes it difficult to draw unequivocal conclusions regarding the functional role of α_1_-ARs in the regulation of retinal vascular resistance and perfusion. While Mori et al. observed a dose-dependent constriction of rat retinal arterioles in response to intravenous administration of noradrenaline, Alm et al. did not observe any effect of exogenously administered noradrenaline on retinal blood flow in cats [84,85]. Dollery et al. reported that intravenously administered noradrenaline decreased retinal vascular diameter in healthy humans, whereas other studies detected only a negligible impact of circulating noradrenaline and the α_1_-adrenergic agonist, phenylephrine, on human retinal vessel diameter and blood flow [86,87,88].

In general, in vivo studies investigating the effects of systemically applied adrenergic agonists or antagonists on retinal vascular responses are hampered by considerable changes in systemic blood pressure induced by these ligands. Due to the pronounced autoregulatory capability of the retinal vascular bed, the resulting changes in ocular perfusion pressure may also induce compensatory responses of the retinal vasculature, which makes it difficult to discern between direct pharmacological effects on retinal vessels and their reactive responses to systemic blood pressure changes [70,71,88,89]. This methodological dilemma may, at least in part, explain the contradictory results of the in vivo studies outlined above.

Ichikawa et al. and Hara et al. reduced systemic influences by intravitreal drug injection and observed constriction of rabbit retinal arteries in response to phenylephrine, which was attenuated after application of the α_1_-AR antagonist, bunazosin [76,90]. Additionally, in vitro studies using isolated eyes, retinal tissue grafts, and preparations of retinal arterioles also demonstrated a constrictive effect of α_1_-adrenergic agonists on retinal blood vessels [71,91,92,93,94].

In this context, Hoste et al. reported that on the one hand, the contractile response of bovine retinal arteries to α_1_-adrenergic stimulation was greatly masked under resting conditions [94]. On the other hand, retinal arteries became sensitive to α_1_-adrenergic agonists when activated by circumferential stretch and displayed an enhanced myogenic vasoconstrictor response to elevated perfusion pressure during α_1_-adrenergic stimulation [94]. In the study of Yu et al., the extent of vasoconstriction in isolated porcine retinal arterioles with intact endothelium in response to α_1_-adrenergic agonists differed between intra- and extra-luminal drug application. Adrenaline and noradrenaline caused concentration-dependent vasoconstriction, which was larger when applied intraluminally [70,92].

Although the aforementioned in vitro studies found a constrictive effect of α_1_-adrenergic agonists on retinal vessels with an intact endothelium, the reported vascular responses were mostly weak and had a relatively high threshold [91,92,93,94]. In a study conducted on murine retinal explants, our group has demonstrated that retinal arteriole constriction to α_1_-adrenergic stimulation is largely masked by endothelial mechanisms and becomes more relevant when the endothelium is damaged [67]. In contrast, a prior study of our own performed on endothelium-intact murine ophthalmic arteries under similar experimental conditions has shown pronounced vasoconstrictive responses to α_1_-adrenergic stimulation [67]. These findings suggest that endothelial modulation of α_1_-adrenergic vasoconstriction differs between retinal and retrobulbar blood vessels.

Due to its characteristic properties, the endothelium of retinal vessels represents a mechanical barrier, which is considered to prevent most blood-borne hydrophilic compounds, including catecholamines, from reaching vascular smooth muscle cells [20,50,62,63,64,70,95]. In the referred study, however, the blood-retinal barrier was circumvented, since vasoactive substances were applied extraluminally [67]. Therefore, the observed attenuating influence of the endothelium most likely results from endothelium-derived vascular mechanisms that functionally antagonize constrictive responses of retinal arterioles to α_1_-adrenergic stimuli. Endothelium-dependent attenuation of retinal α_1_-AR-mediated vasoconstriction appears physiologically plausible, since it would confer a safety net by protecting the retina against inappropriate reductions in blood flow induced by elevated levels of catecholamines during exercise, hemorrhage, or stress. Based on studies in various vascular beds, including cerebral vessels, it is well documented that vascular responses to α_1_-adrenergic stimuli are modulated by the vascular endothelium and are altered when endothelial function is impaired [96,97,98,99,100,101,102,103,104,105,106]. Apparently, the endothelium mitigates the vasoconstrictive impact of elevated circulating catecholamine levels particularly in organs whose uncompromised blood supply and functioning are of vital importance during fight and flight responses [34,39,107]. Conversely, under conditions of endothelial impairment, e.g., in atherosclerosis and diabetes, vascular sensitivity to α_1_-adrenergic vasoconstrictors appears to be increased [107,108,109,110,111,112].

In general, endothelial compensation of α_1_-AR-mediated vasoconstriction is attributed to relaxing factors, such as nitric oxide, released by endothelial cells [113,114,115,116,117,118,119]. While nitric oxide release in response to increased shear stress during vascular constriction is considered a possible endothelial mechanism counteracting vasoconstriction, some other studies suggest that activation of α_1_-ARs located on endothelial cells also promotes the endothelial release of nitric oxide, functionally antagonizing adrenergic vasoconstriction [29,107,120,121,122,123,124,125].

Retinal pathologies, such as diabetic retinopathy, arterial occlusive disease or glaucoma, are associated with impaired endothelial function [12,126,127,128]. However, so far, no compelling evidence has been provided that under retinal pathological conditions α_1_-AR-mediated vasoconstriction becomes a relevant contribution factor, although an in vivo study in rabbits suggested that blockade of α_1_-AR signaling may alleviate the impairment in blood flow and retinal function caused by nitric oxide synthase inhibition [129].

#### 3.1.3. Contribution of Individual α_1_-AR Subtypes to Vascular Reactivity in the Retina

In their in vivo study, Mori et al. aimed to identify the α_1_-AR subtype(s) involved in noradrenaline-induced constriction of retinal arterioles in anaesthetized rats by analyzing the vascular effects of systemically administered subtype-preferring agonists and antagonists. The authors concluded from their results that vascular constriction to noradrenaline in rat retinal arterioles is primarily mediated by the α_1A_- and the α_1D_-AR, and that this finding corresponds to the situation in the rat peripheral circulation [84]. From a methodological point of view, the interpretation of the results is hampered by the confounding influence of retinal vascular autoregulation in an in vivo setting and by the lack of highly selective agonists and antagonists for all α_1_-AR subtypes [24,28,35,70,71,88,89,130,131].

Using an in vitro approach and gene-targeted mice deficient in individual α_1_-AR subtypes, our group has recently demonstrated that α_1_-AR-mediated vasoconstriction in retinal arterioles with damaged endothelium is predominantly conveyed by the α_1B_-AR subtype [67]. By contrast, in the murine ophthalmic artery, which is directly afferent to the retinal vasculature, the α_1A_-AR subtype has previously been shown to mediate constrictive responses to adrenergic stimuli [132]. These results indicate that the retrobulbar (α_1A_-AR) and the retinal (α_1B_-AR) vasculature are under the functional control of different α_1_-AR subtypes. This finding is in line with previous studies reporting that the distribution of individual α_1_-AR subtypes and their contribution to adrenergic vasoconstriction varies considerably between circulatory beds, between different-sized vessels of a given vascular bed, and between different species [24,34,39,133,134,135].

Although mRNA for all three α_1_-AR subtypes was found to be expressed at similar levels in murine retinal arterioles, α_1_-AR-mediated vasoconstriction is predominantly mediated by the α_1B_-AR subtype [67]. Several studies have provided evidence that the presence of mRNA or protein of a particular α_1_-AR subtype does not necessarily go along with its participation in vasoconstriction [31,132,136,137,138]. Furthermore, each receptor subtype can activate distinct downstream signaling components in the G_q/11_ signaling pathways and also couple to other independent signaling pathways [35,36,131,139,140,141]. Therefore, a particular α_1_-AR subtype that does not contribute to vasoconstriction despite its expression may, nevertheless, be involved in the regulation of other physiological or pathophysiological processes of the retinal vasculature. However, whether these results derived from animal models correctly represent the expression and function of α_1_-AR subtypes in the human retinal vasculature remains to be elucidated.

### 3.2. α_2_-ARs

#### 3.2.1. Expression of α_2_-ARs in the Retina

In 1982, Osborne detected retinal α_2_-ARs in the bovine retina in binding studies utilizing [^3^H]noradrenaline and [^3^H]clonidine [142]. In 1986, α_2_-AR binding sites were identified in the inner plexiform layer of the rat retina by [^3^H]para-aminoclonidine ([^3^H]PAC) [77]. Most of these sites were also related to the proximal layer of cell bodies in the inner nuclear layer and with some putative displaced amacrine or ganglion cell bodies [77]. Furthermore, α_2_-ARs were identified in calf retinal cellular membranes by binding experiments employing the radiolabelled antagonists, [^3^H]-RX 781,094 and [^3^H]-rauwolscine [143]. Matsuo and Cynader found α_2_-AR binding sites in the retina of human cadaveric eyes by an in vitro ligand-binding technique and autoradiography [144].

Three distinct subtypes of α_2_-ARs, α_2A_, α_2B_, α_2C_, have been identified by molecular and pharmacological characterization techniques [145]. The International Union of Pharmacology has identified species orthologues and termed them α_2A_-AR in humans and α_2D_-AR in rats and mice [27]. Venkataraman et al. indicated that the *α_2D_-AR* gene in the bovine retina was a structural variant of the rat and mouse genes and defined the α_2D_-AR subtype in the bovine retina [146]. The expression of the α_2D_-AR subtype was detected in the bovine retina and its photoreceptors [146]. Messenger RNA of the α_2D_-AR was identified in the bovine retina by amplification through reverse transcription-polymerase chain reaction (RT-PCR) [146]. It is now generally accepted that the α_2D_-AR represents a species variant of the α_2A_-AR [147].

Immunohistochemical studies revealed the presence of α_2_-ARs (specifically α_2A_-ARs) on rat RGCs and inner nuclear layer cells [148]. In the human retina, Kalapesi et al. found α_2A_-ARs on human RGCs and cells in the inner and outer nuclear layers [148].

In addition, Pfizer et al. reported that in rat retinas, α_2A_-ARs were mainly found in the cell bodies located in the ganglion cell layer, the inner plexiform layer and the outer plexiform layer [149]. In other immunohistochemical studies, α_2A_-AR staining was also observed in the membrane of cells located in the inner nuclear layer, specifically amacrine and horizontal cells. In human and monkey retinas, the α_2A_-AR staining pattern was relatively consistent with that observed in rats [149]. In contrast to α_2A_-ARs, α_2B_-AR immunoreactivity was observed in all retinal layers, especially in the presynaptic regions of neurons. In the outer retina, α_2B_-AR immunoreactivity was seen in more than one cell type, such as the inner segments of photoreceptors, Müller cells, and bipolar cells [149]. Moreover, Pfizer et al. observed that α_2A_-ARs were mainly present in the cell membrane of photoreceptor cells and in their inner segments [149]. Notably, some studies reported that many commercially available antibodies directed against ARs, including α_2_-ARs, lack sufficient specificity [80,82,83]. Based on these findings, expression studies employing commercially available AR antibodies need to be interpreted with caution [82].

#### 3.2.2. α_2_-AR in Retinal Neuroprotection

The family of α_2_-ARs is one of the pharmacological targets of the natural stress hormone, norepinephrine, and is involved in modulating cellular resistance and adaptation to stress stimuli [150]. α_2_-ARs were first described as presynaptic receptors inhibiting the release of various transmitters from neurons in the central and peripheral nervous systems [151]. In vivo studies revealed that α_2_-AR stimulation reduces ischemic injury in the brain [152]. This effect has been largely attributed to its classic presynaptic inhibition of signaling molecules released by blocking Ca^2+^ channels, activating K^+^ channels, or reducing active release sites [153,154]. Neuroprotective treatment strategies for retinal diseases whose course includes neuronal degeneration have arisen a great deal of interest. Some studies have assessed the functional role of α_2_-ARs in the retina in a variety of animal models through the mechanism of α_2_-mediated neuroprotection. For example, Donello et al. suggested that activation of α_2_-ARs might reduce ischemic retinal injury and preserve retinal function following transient ischemia by preventing extracellular glutamate and aspartate accumulation [155]. An in vivo study by Manuel et al. revealed neuroprotective effects of α_2_-ARs in preventing RGC death after transient retinal ischemia [156]. In the study, pre-treatment with two α_2_-AR-selective agonists, AGN 191,103 and brimonidine tartrate, has proven very effective not only in preventing rapid RGCs loss, but also long-term RGCs loss in a rat model [156].

Glaucoma with elevated intraocular pressure (IOP) often continues to progress even after the reduction of IOP to normal levels [5]. The progressive loss of vision in eyes with glaucoma is a result of RGCs degeneration, emphasizing the need for a neuroprotective therapy [157]. The α_2_-AR agonist, brimonidine, and other α_2_-AR agonists were shown to lower IOP mainly by reducing aqueous humor production and by increasing uveoscleral outflow [158,159,160]. In a rat model of chronically elevated IOP, pharmacological activation of α_2_-ARs exerted neuroprotective effects in RGCs, irrespective of the IOP level [157]. In addition, the α_2_-AR agonist, brimonidine, preserved visual function in glaucoma patients with low/normal IOP and high IOP, suggesting that pharmacological α_2_-AR activation may exert neuroprotective effects IOP independently [161,162,163]. Various mechanisms underlying the neuroprotective activity of α_2_-AR agonists have been proposed. For example, activation of α_2_-ARs was shown to decrease the IOP-induced overexpression of intermediate filament glial fibrillary acidic protein (GFAP) in Müller cells, suggesting that activation of α_2_-ARs may reduce stress responses in glial cells. [157]. Furthermore, the α_2_-AR agonist, brimonidine, was reported to protect RGCs from the effects of chronic ocular hypertension through mechanisms involving α_2_-AR-mediated survival signal activation and up-regulation of endogenous neurotrophic factors in the rat retina [164]. Another study demonstrated that α_2_-adrenergic modulation of *N*-methyl-d-aspartate (NMDA) receptor function was an important mechanism for neuroprotection in experimental glaucoma models [154]. Brimonidine may protect RGCs by preventing abnormal elevation of cytosolic free Ca^2+^ evoked by NMDA receptors in RGCs under stress conditions [165,166]. Other studies suggested that brimonidine-mediated inhibition of the cyclic adenosine 3′,5′-monophosphate/protein kinase A (cAMP/PKA) pathway could be an important mechanism to protect RGCs from glaucomatous neurodegeneration [162]. Based on a recent study in a rat glaucoma model, Zhou et al. suggested that α_2_-AR activation hyperpolarizes RGCs by improving the γ-aminobutyric acid (GABA) receptor response to spontaneous and elicited GABA release, thus reducing the risk for excitotoxicity and RGC injury [167].

Activation of α_2_-ARs has also been shown to exert neuroprotective effects in other retinal diseases, such as light-induced photoreceptor damage, retinal detachment, and optic nerve injury [168,169,170]. For example, α_2_-adrenergic agonists were shown to induce the expression of basic fibroblast growth factor (bFGF) in photoreceptors in vivo and to alleviate light-induced damage in the retina [168]. In retinal detachment, stimulation of α_2_-AR signaling protected photoreceptors by inhibiting oxidative stress and inflammation [170]. Furthermore, in a mouse optic neuritis model, topical administration of the α_2_-AR agonist, brimonidine, exerted neuroprotective effects against RGCs death by increasing retinal bFGF expression in the retina, particularly in Müller cells and RGCs [171]. In contrast, in mechanic optic nerve injury, activation of α_2_-AR signaling promoted optic nerve regeneration via activation of extracellular-signal-regulated kinase (ERK) phosphorylation [169].

Activation of α_2_-ARs was suggested to induce vasoconstriction in the vasculature distal to the ophthalmic artery, such as ciliary and retinal blood vessels [172,173,174]. While the α_2A_-AR subtype was proposed to mediate adrenergic vasoconstriction in porcine ciliary arteries, no suggestion regarding the contribution of individual α_2_-AR subtypes has been made for retinal blood vessels [172].

Only little is known about the functional role of individual α_2_-AR subtypes in the retina. A study by Harun-Or-Rashid et al. showed that α_2A_-ARs are expressed by chicken Müller cells and that pharmacological activation of the α_2A_-AR subtype triggers a mitogen-activated protein kinase (MAPK)-dependent response with phosphorylation of ERK1/2 both in vivo and in vitro. Taken together, most studies indicate that activation of α_2_-ARs raises neuronal resistance to retinal injury suggesting that the receptors may become potent treatment targets in various retinal diseases. However, the role of individual α_2_-AR subtypes remains to be better characterized in the retina.

### 3.3. β-ARs

#### 3.3.1. Expression of β-ARs in the Retina

In 1986, Zarbin et al. localized β-AR binding sites by [^3^H]dihydroalprenolol in nearly all rat retinal layers (the outer nuclear layer, outer plexiform layer, inner nuclear layer, and inner plexiform layer), but with the lowest concentration in the outer nuclear layer [77]. In the human retina, β-AR binding sites were visualized by [125I] (-) iodocyanopindolol in vitro autoradiography [175].

Three distinct β-AR subtypes, β_1_-AR, β_2_-AR and β_3_-AR, have been identified [176]. In 2001, the β_4_-AR was hypothesized by Granneman as a novel state of the β_1_-AR [176]. Six years later, the β_4_-AR was considered by Madamanchi as a low-affinity state of the β_1_-AR, which still needs pharmacological and genetic characterization [177]. Among β-AR subtypes, β_1_-AR and β_2_-AR binding sites were found in bovine retinal vessels and in the neural retina [178]. The presence of functional β_3_-ARs has also been verified in rat retinal blood vessels [179]. By immunohistochemistry, β_3_-ARs were localized in the inner capillary plexus of the mouse mid-peripheral retina, whereas β_1_-ARs and β_2_-ARs were localized in rat cultured Müller cells [180]. However, since many commercially available antibodies lack sufficient specificity for β-ARs, data based on antibodies need to be interpreted with caution [83]. Expression of β_3_-ARs has been shown for the first time on cultured human retinal endothelial cells in 2003 [181]. In that study, pharmacological activation of β_3_-ARs promoted migration and proliferation of endothelial cells [181]. Based on the wide distribution of β-ARs in retinal blood vessels and the neural retina, β-ARs are believed to play an important role in retinal vascular and neuronal function. The localization of individual AR subfamilies in individual retinal layers and structures is shown in Figure 2.

#### 3.3.2. Role of β-ARs in the Retina

Stress conditions, such as hypoxia, can cause catecholaminergic overstimulation in the cardiovascular system, which in turn may activate β-ARs [182]. A study in mice reported that the noradrenaline level increased by approximately 90% in the hypoxic retina compared to normoxic conditions [183]. Activation of β-ARs is considered to upregulate the hypoxia-inducible factor-1α (HIF-1α) and vascular endothelial growth factor (VEGF), which plays a key role in the formation of pathogenic blood vessels in various retinal diseases, such as retinopathy of prematurity and diabetic retinopathy [183,184,185]. The β-AR blocker, propranolol, effectively inhibited the increase of VEGF expression caused by hypoxia and the consecutive neovascular response in the retina [55]. Likewise, propranolol administered subcutaneously reduced VEGF and HIF-1α levels in an oxygen-induced retinopathy (OIR) mouse model, suggesting that β-AR blockade was protective against retinal angiogenesis and ameliorated blood-retinal barrier dysfunction [186].

Intriguingly, a novel β-AR agonist, compound 49b, was reported to decrease VEGF levels in the diabetic rat retina [187]. The effect of compound 49b was attributed to an increase in insulin-like growth factor binding protein 3 (IGFBP-3), which reduced VEGF levels via modulation of eNOS and PKC pathways [187]. These seemingly contradictory findings regarding the impact of β-AR agonists and antagonists on VEGF levels suggest that the effects may be mediated through diverse regulation mechanisms depending on the retinal disease and the experimental setting.

β-AR activation was also shown to increase human and mouse pericyte survival under diabetic conditions [188]. Conversely, a significant decrease in the number of pericytes has been reported in the rat retina after surgical removal of the superior cervical ganglion, which supplies the eye with sympathetic nerve fibers [189]. These findings suggest that proper β-AR signaling is essential for pericyte survival. An in vitro study proposed that β-ARs are involved in the regulation of inducible nitric oxide synthase (iNOS) expression [190]. Activation of β-ARs reduced levels of iNOS and other inflammatory molecules, such as interleukin (IL)-1β, tumor necrosis factor-α (TNF-α), and prostaglandin E2 (PGE2) in human retinal endothelial cells and rat Müller cells in an in vitro model of hyperglycemia [191]. The proposed mechanism for the protective effects was that the stimulating β-ARs decrease the levels of the MAPK family members, PKA, p38 MAPK, and p42/p44 MAPK, in human retinal endothelial cells [190].

Stimulation of β-AR by agonists can activate members of the G protein-coupled receptor (GPCR) kinase (GRK) family, which is the potential mechanism that leads to β-AR desensitization [192]. A first mechanism underlying desensitization is phosphorylation of the GPCR [193]. After coupling to activated receptors, G proteins are phosphorylated by GRKs [194,195]. Seven GRKs (GRK1–7) have been identified, at present [196]. GRK1, as a rhodopsin kinase, is responsible for phosphorylating rhodopsin, which is richly expressed in the retinal rod and cone cells [197]. β-ARs can be phosphorylated by a protein kinase termed GRK2 [198]. The GRK-phosphorylated receptor binds to arrestins, leading to the uncoupling of the receptor from the G protein, desensitizing the agonist-induced response, and subsequently mediating the internalization of receptors [195,199]. Consequently, the GRK-arrestin pathway plays a central role in the desensitization of GPCR responses [195,199]. Two arrestin subtypes were found to be expressed in the retina, arrestin-1 and -4. These arrestins are specialized in binding light-activated phosphorylated rhodopsin and suppressing G protein activation [200,201]. It has been demonstrated that retinal and nonretinal arrestins mediate suppression of GPCRs, suggesting a common mechanism for desensitizing ARs [202]. Apart from this role, arrestins were also shown to be involved in receptor-mediated endocytosis by clathrin-coated pits [202,203].

#### 3.3.3. Contribution of Individual Receptor Subtypes to β-Adrenergic Function in the Retina

Hypoxia was reported to trigger the release of catecholamines, which have been shown to contribute to the increase in retinal VEGF expression, causing pathologic neovascularization [183,184,185]. In a mouse model of OIR, deletion of β_1_- and β_2_-ARs reduced retinal VEGF receptor-2 expression and abolished the development of vascular abnormalities in the superficial plexus of the retina [204]. In another study employing a mouse model of OIR, β_1_- and β_2_-AR blockade by propranolol was shown to reduce the expression of VEGF and to ameliorate retinal dysfunction [186]. Other studies demonstrated that ICI 118,551, a selective β_2_-AR blocker, decreased retinal levels of proangiogenic factors and reduced pathogenic neovascularization in a mouse OIR model, suggesting that β_2_-AR blockade may be effective in the blockade of retinal angiogenesis [205].

As we mentioned before, there are inconsistent results regarding the role of β-AR activation or blockade on VEGF expression and pathogenic vessel formation. While most studies reported on inhibitory effects of pharmacological β-AR blockade on VEGF formation some other studies observed blockade of VEGF formation by β-AR agonist exposure [55,182,183,184,185,186]. An explanation to these seemingly contradictory results provides a study by Dal Monte et al., which suggested that the nonselective β-ARs agonist, isoproterenol, can cause agonist-induced β_2_-AR desensitization that downregulates the expression of β_2_-ARs in the retina, which in turn exerts a downregulation effect on VEGF expression in OIR [183]. Jiang et al. proposed that β_2_-AR KO mice exhibited certain features similar to diabetic retinopathy, resulting in retinal cell death [206]. A study in β_2_-AR KO mice has shown a functional link between β-ARs and insulin receptor signaling pathways in the retina [200]. Furthermore, β_2_-ARs can maintain insulin receptor signaling in retinal Müller cells, which potentially supports neuroprotective effects promoted by β-AR stimulation in diabetic retinopathy models [200]. Xamoterol, a β_1_-AR agonist, attenuated iNOS expression in human retinal endothelial cells grown in high glucose medium [191]. Studies by Mori et al. have demonstrated that stimulation of β_1_-, β_2_-, and β_3_-ARs can cause dilation of retinal arterioles in rats, thus, increasing retinal blood flow [207,208]. Moreover, the latest study by Mori et al. reported that retinal vasodilation by β_2_-AR stimulation is mediated via a G_i_ protein by activation of large-conductance Ca^2+^-activated K^+^ channels [209].

β_3_-ARs were shown to be involved in the neovascularization processes of various retinal vascular diseases [186]. For example, β_3_-ARs were upregulated in response to hypoxia in an OIR mouse model with dense β_3_-ARs immunoreactivity in engorged retinal tufts, suggesting that activation of β_3_-ARs is likely to constitute an important part in pathologic angiogenesis [186]. It has also been demonstrated that the β_3_-AR antagonists, L-748,337 and SR59230A, downregulated retinal VEGF release in hypoxia via modulation of the nitric NO signaling pathway [210]. Moreover, the β_3_-AR agonist, CL316243, was shown to reduce retinal damage following intravitreal injection of N-methyl-D-aspartate (NMDA) in rats [211]. Furthermore, β_3_-ARs, which differ from β_2_-ARs with regard to a lack of GRK, are resistant to agonist-induced desensitization [192,212]. Taken together, based on the studies performed so far, the β_3_-AR appears to be an attractive therapeutic target for the treatment of ischemic retinal diseases.

Notably, β-AR genes have genetic polymorphisms caused by mutations in the gene promoter, leading to changes in the expression of receptors and the regulation of signal transduction [195,213,214]. The mutations of ARs mainly affect receptor responses and are associated with some diseases [214]. An in vivo study tested the effects of two polymorphisms (codon 16 and codon 27) of the β_2_-AR on agonist-mediated vascular desensitization, suggesting that the arginine at position 16 (Arg16) polymorphism (the substitution of glycine for arginine) of the β_2_-AR is associated with enhanced agonist-mediated desensitization in the vasculature [215]. However, to the best of our knowledge, no association between genetic polymorphisms of β-AR genes and retinal diseases has been described, so far.

## 4. Future Directions and Clinical Implications

Although α_1_-ARs play a potential physiological and/or pathophysiological role in the regulation of retinal vascular tone, retinal α_1_-adrenergic vasoconstriction is largely masked by endothelial mechanisms under physiological conditions and becomes more relevant when the endothelium is damaged. Therefore, α_1_-AR signaling pathways may represent therapeutic targets primarily in the context of retinal pathologies associated with impaired endothelial function. However, there are inconsistent findings regarding the subtype mediating adrenergic vasoconstriction in retinal blood vessels. While Mori et al. found that α_1A_- and α_1D_-ARs mediated vasoconstriction in vivo in the rat retina, Gericke et al. reported that the α_1B_-AR-mediated adrenergic vasoconstriction in the isolated mouse retina [67,84]. It remains to be determined which α_1_-AR subtype mediates vascular responses in the human retina. However, the lack of highly specific pharmacological ligands and antibodies for individual α_1_-AR subtypes hampered the progress in this research area so far. From a clinical point of view, subtype-selective agonists and antagonists would constitute a therapeutic approach to specifically influence retinal perfusion.

It should be emphasized, however, that many issues remain poorly understood and need further investigation. The location of α_1_-ARs in general and of the three α_1_-AR subtypes in particular within the architecture of retinal vessels is still elusive but may be indicative of their vascular function(s) and relevant for their pharmacological accessibility. The endothelial mechanisms involved in masking retinal α_1_-adrenergic vasoconstriction have not yet been identified. In retinal diseases associated with endothelial dysfunction, the actual pathogenic contribution of α_1_-AR-mediated vasoconstriction remains an open question.

Various animal experiments and cell culture studies revealed neuroprotective effects of the selective α_2_-AR agonist, brimonidine, in the retina [148,150,216]. In 2011, a randomized clinical trial reported on the neuroprotective effects of topically applied brimonidine tartrate 0.2% in preventing visual field loss progression in patients with low-pressure glaucoma, which supports the concept of direct activation of retinal α_2_-ARs [163]. In contrast, another trial failed to show neuroprotective effects of 0.2% brimonidine tartrate in patients with non-arteritic anterior ischemic optic neuropathy [217]. In a pilot study on patients with retinal dystrophies, topical treatment with brimonidine suggested a trend towards reduced disease progression [218]. However, the number of patients (n = 26) was relatively small and the mean follow-up period (mean 29 months) relatively short, which does not allow to draw unequivocal conclusions. Taken together, the retinal neuroprotective effects of brimonidine obtained in various experimental disease models remain to be confirmed in large human trials.

Propranolol, which has a high affinity to β_1_- and β_2_-AR subtypes, is the only β-AR antagonist that has been tested in clinical trials, so far [55]. Propranolol 0.1% eye micro-drops have been developed and administered in a multicenter pilot clinical trial to analyze the safety and efficacy in treating preterm newborns with stage 2 ROP [219]. However, the second stage of this study was discontinued, since the fourth of the 19 newborns showed a progression to stage 2 or 3 with additional disease [219]. Based on animal studies, β_3_-ARs appear to be involved in pathogenic vessel formation in the ischemic retina. Hence, future studies are needed to explore β_3_-AR ligands in human ischemic retinal diseases.

## 5. Conclusions

In conclusion, the studies reviewed in this article provide evidence for the presence of α_1_-, α_2_- and β-ARs in the retina of various species, including humans. α_1_-ARs function as stimulatory receptors involved mainly in the contraction of vascular smooth muscle, resulting in vasoconstriction. In contrast, α_2_-ARs are primarily expressed in neurons and glia. Its pharmacological activation was shown to lower IOP and to induce neuroprotective effects in the retina. β-ARs are expressed in blood vessels, neurons and glial cells, where they were shown to contribute to the regulation of vascular diameter and to responses to hypoxia. The results of numerous in vitro and in vivo studies suggest that these retinal receptors are functionally active and, hence, may constitute potential targets for the treatment of several retinal diseases.

## Figures and Tables

**Figure 1 cells-09-02594-f001:**
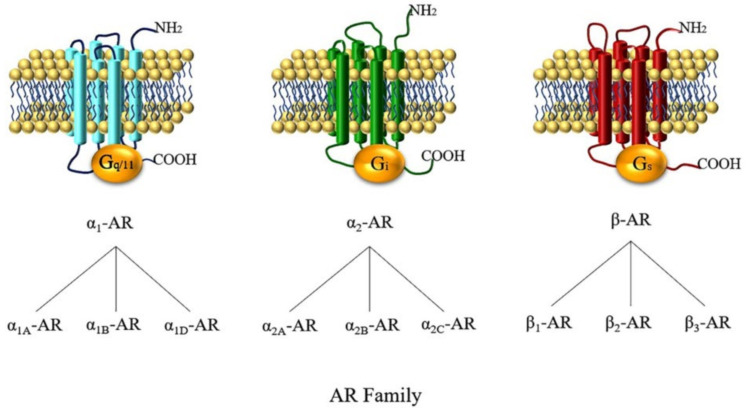
The three adrenoceptor (AR) subfamilies and their subtypes. α_1_-, α_2_-, and β-ARs mainly couple to G_q/11_, G_i_, and G_s_ proteins, respectively. To be more exact, the α_2A_-AR subtype has an unusual dual pharmacological effect by coupling to G_i_ proteins when concentrations of agonists are low and mainly coupling to G_s_ proteins when concentrations are high [43]. β-ARs are mainly coupled to G_s_ proteins and both β_1_- and β_2_-AR are able to switch their G protein-coupling specificity from G_s_ to G_i_ proteins [44].

**Figure 2 cells-09-02594-f002:**
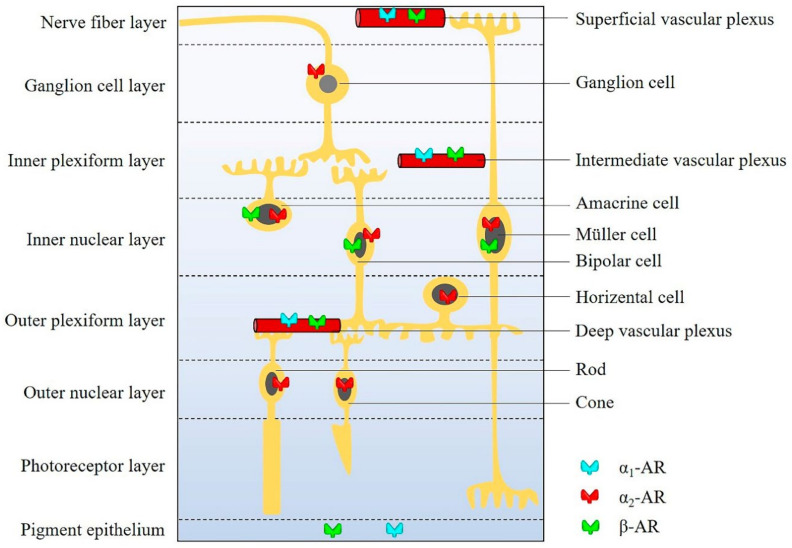
The distribution of ARs in individual retinal layers and structures.

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
