# Peer review of "The Role of Adrenoceptors in the Retina"

_cells, 2020, doi:10.3390/cells9122594_

Round 1

Reviewer 1 Report

In this literature review, the authors describe work relating to adrenoreceptors found in the retina. The work is thorough and scholarly, but more attention should be made to readability and assimilating the information into a digestible form. I recommend the following changes before accepting for publication:

1. This review needs reorganization. For a review focused on adrenoreceptors, adrenoreceptors are only first mentioned in the fifth paragraph. Much of what precedes this paragraph can be removed without impacting the review.

2. The review should be edited to remove superfluous details. A shorter and more focused review will better emphasize the role of adrenoreceptors in the retina. For example, details of the clinical trials of brimonidine and propranolol can be substantially reduced. Additional subheadings will help organize the review better.

3. More information should be provided comparing the functions of the different classes of adrenoreceptors, especially as the review is structured so that each class is examined in turn. In the absence of this information, it is difficult to understand the importance of each class and how this might relate to their location in the retina.

4. Additional figures or tables should be provided showing the location of the adrenoreceptors in major parts of the retina. This will improve the understanding of the review for readers less familiar with retina anatomy and will help summarize the large amounts of data presented.

5. The effect of pharmacological drugs on various adrenoreceptors should be placed in a separate section.

Minor comments

1. Where there are contradictions in the literature, more effort should be spent to discuss the results rather than just presenting the conflicting evidence.

2. Generalized statements such as “cellular components” (line 60) and “various species” (line 122) should be avoided and replaced with specifics.

3. “Moreover” is overused (18 instances).

Author Response

Responses to the Reviewers’ Comments

Reviewer 1

  1. This review needs reorganization. For a review focused on adrenoreceptors, adrenoreceptors are only first mentioned in the fifth paragraph. Much of what precedes this paragraph can be removed without impacting the review.

To 1.) According to the Reviewer’s suggestion, we reorganized the introduction and removed some dispensable sentences preceding the description of adrenoceptors.

  1. The review should be edited to remove superfluous details. A shorter and more focused review will better emphasize the role of adrenoreceptors in the retina. For example, details of the clinical trials of brimonidine and propranolol can be substantially reduced. Additional subheadings will help organize the review better.

To 2.) We removed some details, including the details of the clinical trials of brimonidine and propranolol. We also added additional subheadings.

  1. More information should be provided comparing the functions of the different classes of adrenoreceptors, especially as the review is structured so that each class is examined in turn. In the absence of this information, it is difficult to understand the importance of each class and how this might relate to their location in the retina.

To 3.) We added more information comparing the functions of the different classes of adrenoceptors in the conclusion (page12, line 496-499). In addition, we depicted the location of each AR class within the retina (Figure 2).

  1. Additional figures or tables should be provided showing the location of the adrenoreceptors in major parts of the retina. This will improve the understanding of the review for readers less familiar with retina anatomy and will help summarize the large amounts of data presented.

To 4.) We added one additional figure (Figure 2) to show the location of the adrenoreceptors the retina.

  1. The effect of pharmacological drugs on various adrenoreceptors should be placed in a separate section.

To 5.) We have chosen to report on some studies using pharmacological drugs on various adrenoreceptors in the sections on individual receptors. In the section entitled “Future directions and clinical implications” we placed the clinically important drugs of all receptor classes to help the reader to extract the clinically important information.

Minor comments

  1. Where there are contradictions in the literature, more effort should be spent to discuss the results rather than just presenting the conflicting evidence.

To 1.) We now discuss some of the mechanism underlying the contradictions (lines 375-377, lines 389-404, lines 415-422).

  1. Generalized statements such as “cellular components” (line 60) and “various species” (line 122) should be avoided and replaced with specifics.

To 2.) We have changed “various species” to “various mammalian species” (page 3, line104) and “cellular components” to “cardiovascular system” (page2,line44)

  1. “Moreover” is overused (18 instances).

To 3.) We have reduced “moreover” and use other words instead (underlined).

Reviewer 2 Report

This is a very interesting, timely, and comprehensive review article on retinal adrenergic receptors. However, a couple of points that should have been covered have been missed by the authors, so below are a couple of suggestions to help them improve their manuscript:

1) What is known about desensitization/downregulation and, more broadly, regulation of retinal adrenoceptors? GRKs and arrestins play crucial roles in regulation of rhodopsin/opsins in the retina, as well as of adrenergic receptors outside the retina (e.g. see: Prog Mol Biol Transl Sci. 2018;159:27-57; Pharmacogenomics. 2012;13:323-41; Prog Mol Biol Transl Sci. 2018;160:47-61; for pertinent reviews). Therefore, a brief discussion of visual/retinal GRKs & arrestins, as they pertain to retinal adrenoceptor function, must be included. 

2) A brief section on what is known regarding human adrenoceptor polymorphisms and their impact on receptor function in the retina should be included, as well. 

Author Response

Responses to the Reviewers’ Comments

Reviewer 2

  1. What is known about desensitization/downregulation and, more broadly, regulation of retinal adrenoceptors? GRKs and arrestins play crucial roles in regulation of rhodopsin/opsins in the retina, as well as of adrenergic receptors outside the retina (e.g. see: Prog Mol Biol Transl Sci. 2018;159:27-57; Pharmacogenomics. 2012;13:323-41; Prog Mol Biol Transl Sci. 2018;160:47-61; for pertinent reviews). Therefore, a brief discussion of visual/retinal GRKs & arrestins, as they pertain to retinal adrenoceptor function, must be included.

To 1.) We have discussed a brief discussion of visual/retinal GRKs & arrestins, added the reference: Prog Mol Biol Transl Sci. 2018;159:27-57; Pharmacogenomics. 2012;13:323-41; Prog Mol Biol Transl Sci. 2018;160:47-61 (page10, line389-404).

  1. A brief section on what is known regarding human adrenoceptor polymorphisms and their impact on receptor function in the retina should be included, as well.

To 2.) We have added a section on adrenoceptor polymorphisms and their impact on receptor function in the retina (page11, line444-452).

Round 2

Reviewer 2 Report

I applaud the authors for the great job they have done that has improved their manuscript significantly. Their review article is now quite comprehensive and reads very well. I have no further comments, except to direct the authors` attention to the references of their revised manuscript: I think some references appear in duplicate; please re-check your reference list carefully and remove any duplicate citations.